# Cross-sectional comparisons of sodium content in processed meat and fish products among five countries: potential for feasible targets and reformulation

Yuzhu Song ,[1] Yuan Li,[2,3] Chunlei Guo,[2] Yishan Wang,[4] Liping Huang,[5] Monique Tan ,[6] Feng J He,[7] Terry Harris,[8] Graham A MacGregor,[7] Jingmin Ding,[2] Le Dong,[2] Yu Liu,[9] Huijun Wang,[10] Puhong Zhang,[3,11] Yuxia Ma[1]

For numbered affiliations see end of article.

**Correspondence to**
Professor Yuxia Ma;
mayuxia@hebmu.edu.cn and
Dr Puhong Zhang;
zpuhong@georgeinstitute.org.cn

## ABSTRACT

**Introduction** Reducing sodium intake has been identified as a highly cost-effective strategy to prevent and control high blood pressure and reduce cardiovascular mortality. This study aims to compare the sodium content in processed meat and fish products among five countries, which will contribute to the evidence-base for feasible strategies of sodium reduction in such products.

**Methods** Sodium content on product labels of 26 500 prepackaged products, 19 601 meat and 6899 fish, was collected in supermarkets from five countries using the FoodSwitch mobile application from 2012 to 2018. To be specific, it was 1898 products in China, 885 in the UK, 5673 in Australia, 946 in South Africa and 17 098 in the USA. Cross-sectional comparisons of sodium levels and proportions meeting 2017 UK sodium reduction targets were conducted using Kruskal-Wallis H and the $\chi^2$ test, respectively across the five countries.

**Results** The results showed that processed meat and fish products combined in China had the highest sodium level (median 1050 mg/100 g, IQR: 774–1473), followed by the USA, South Africa, Australia, with the lowest levels found in UK (432 mg/100 g, IQR: 236–786) (p<0.001). Similar variations, that is, a twofold to threefold difference of sodium content between the highest and the lowest countries were found among processed meat and fish products separately. Large sodium content variations were also found in certain specific food subcategories across the five countries, as well as across different food subcategories within each country.

**Conclusion** Processed meat and fish products differ greatly in sodium content across different countries and across different food subcategories. This indicates great potential for food producers to reformulate the products in sodium content, as well as for consumers to select less salted food.

## INTRODUCTION

High sodium intake is the major cause of high blood pressure and increases the risk of cardiovascular disease, renal disease and premature mortality.[1] Processed meat and fish products constitute important categories

### Strengths and limitations of this study

► This is the first cross-sectional study to compare the sodium content of processed meat and fish products among five countries.

► Products were obtained only in selected stores at a specific time point in each country.

► We did not capture food-purchasing data to quantify actual sodium consumption of processed meat and fish products.

► The data collection time of different countries is inconsistent. During this period, due to the growing interest in reducing salt policies on a global scale, product reformulation may have changed.

of processed food, which provide high-quality protein, minerals and vitamins to daily diet. The processing itself offers an opportunity to add savory flavour to food and prolong the shelf-life of food products to improve food safety. However, the high sodium content, which is known to be a key factor for quality and sensory attributes of processed meat and fish, otherwise raises a huge public health concern. The global average sodium intake was about 4000 mg/day in 2010, twice the maximum 2000 mg/day recommended by the WHO.[2] A previous study conducted in 2013 in China reported that the average sodium content of processed meat and fish products was 1029 mg/100 g and 1424 mg/100 g, respectively, above half of the recommended daily sodium intake.[3] In developing countries like China, sodium intake mainly derives from cooking, yet with the rapid urbanisation and dietary transition, the consumption of hidden sodium in processed foods including meat and fish products tends to be increasing rapidly.[4] In developed countries, where more than three quarters of sodium intake comes from processed foods, it was estimated that

sodium intake from meat and meat products contributed approximately 16%–25% of total daily sodium intake.[5] In response to the WHO goal of 30% sodium reduction by 2025, various sodium reduction actions have been taken worldwide. It is worth paying attention to the high sodium content of processed meat and fish products.[6]

Many countries have made efforts to reduce the sodium content of processed foods. The UK, the USA and Australia have set voluntary targets for sodium reduction in various categories of processed foods.[7–9] South Africa was the first to include the statutory maximum sodium targets in several processed food categories.[10] This target-based approach has been shown to be effective in reducing sodium content for many food products.[11 12] Within the same food category, the sodium level is much lower in food products in countries with sodium reduction targets than those without the target,[4] which can be demonstrated by one in UK versus China: the median sodium content was on average 4.4-fold less in UK sauces compared with their Chinese equivalents.[13]

The George Institute for Global Health established a global food composition database in 2010 as part of The International Network for Food and Obesity/noncommunicable diseases Research, Monitoring and Action Support (INFORMAS), with an aim to collate and track the nutritional compositions of processed foods worldwide.[14] The global food composition database uses a standardised methodology for data collection and processing, with data available from more than 10 countries as of 2020.[15–17] This makes the comparison of sodium content across countries possible. The five countries cover three developed and two developing countries, which allow the comparison meaningful to instruct sodium reduction among countries especially for developing countries. In addition, the selected countries have their own sodium reduction strategies. The comparison results may provide meaningful implication for sodium reduction through prepackaged food in other countries.

In this study, levels of salt content of processed meat and fish products are compared among five INFORMAS member countries: UK, US, Australia, China and South Africa. These five countries have different sodium reduction strategies and relatively large dataset available for sodium content comparison for processed meat and fish products, which allows for the comparison conductible and meaningful. The purpose of this study is to compare the sodium content level and achievements in sodium reduction for meat and fish products among the five countries, and indicate possible strategies on sodium reduction for different countries.

## MATERIALS AND METHODS
### Data collection
Images of prepackaged foods were taken using smartphone applications (The George Institute Data Collector and FoodSwitch)[15] by trained data collectors as well as consumers through crowdsourcing and uploaded to a central content management system. The information displayed on the packages, including product, nutrition and ingredient information, was then entered into a uniform web-based data management system by professionally trained clerks. All entered information was reviewed by a second data entry clerk for accuracy. Products with verified information were classified according to a standard food categorisation system. This study used data of processed meat and fish products collected in the UK, Australia, South Africa and China available within the George Institute global food composition database, with the data collection time ranging from 2012 to 2018. We also obtained data on processed meat and fish products from the US through Label Insight for non-profit research.

### Data categorisation
In the food categorisation system, processed meat products and processed fish products fall into two independent categories. Processed meat products were further classified to 16 subcategories: meat alternative products, bacon, canned meat, frozen meat, meat burgers, salami and cured meats, sausage and hot dogs, sliced meat, dried meat, pate and meat spreads, kebabs, other meat products, raw flavoured meats, whole hams and similar products, roasted chicken, and raw unflavoured meats. Processed fish products were divided into four subcategories: canned fish, chilled fish, frozen fish and other fish.

### Data exclusion criteria
Products with no declaration of neither sodium nor salt values were excluded. In the case of identical products with the same sodium content, but available in different package sizes, these were regarded as duplicates and only one product was included.

### Data analysis
Sodium value data were obtained from the Nutrition Information Panel. For products with only salt values available, sodium values were calculated from salt values divided by 2.5. Median and IQR were used to describe the distribution of sodium values (mg/100 g) given the non-normal distribution of the data. The Kruskal-Wallis H test was used to compare differences in sodium values of processed meat and fish products across the five countries. If the difference was statistically significant, post hoc tests were carried out using Bonferroni correction. The subcategory with data records equal to or less than 5 was excluded from the analysis for subcategory comparisons.

In reference to the 'Traffic Light' criteria developed by the UK, sodium level was defined as low ($<120$ mg/100 g), medium ($120 \leq$ sodium $\leq 600$ mg/100 g) and high ($>600$ mg/100 g); and expressed as green, amber and red accordingly to a horizontal bar chart to show the sodium contents visually.[18] The 2017 UK sodium reduction targets were used to assess the percentage of products reaching the targets across the five countries.[19] The maximum sodium targets of each category were selected

for ease of comparison, and the average targets were used where maximum targets were not provided. The $\chi^2$ tests were used to compare the proportion of products that meet the 2017 UK sodium reduction targets.

To measure the sodium burden caused by consumption of processed meat and fish products, a sodium intake contribution value was calculated for each category of food products. It was a ratio of daily sodium intake from 100 g product against the WHO maximum sodium recommendation (2000 mg/day), assuming the consumption of processed meat and fish food products for a person were 100 g/day. For each category of the food products, the contribution value was calculated as median sodium content (mg/100 g)/2000 (mg/day)*100% and was highlighted as red, yellow and green, respectively, to represent high (>66%), medium (>33%, ≤66%) and low (≤33%) sodium intake contribution.

A two-sided p value of <0.05 was considered significant in the statistical tests. The analyses were conducted using Stata/SE V.14.2 and IBM SPSS V.21.0.

### Patient and public involvement
No patient involved.

## RESULTS
A total of 33 955 processed meat and fish products were collected from the five countries, of which 7455 (21.96%) were excluded because of missing sodium data or duplicate products, leaving 26 500 (78.04%) products for analysis in this study (figure 1). The total number of products per country ranged from 885 for the UK to 17 098 for the USA (table 1). The number of products per category ranged from 1 in meat alternative products, kebabs and roasted chicken to 2817 in sausages and hot dogs.

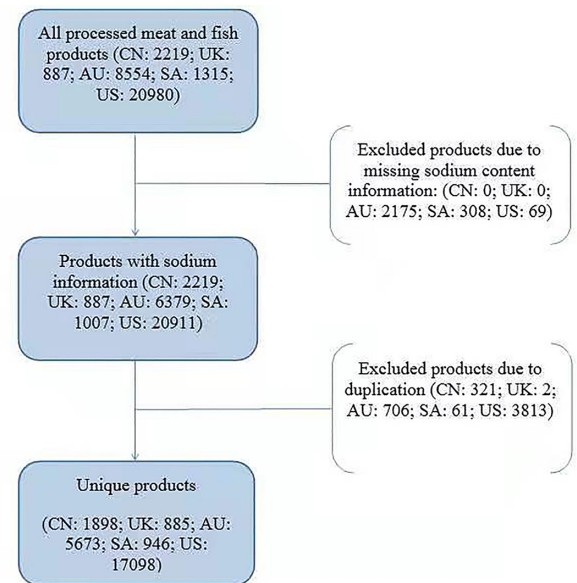

**Figure 1** Flow diagram of product selection. AU, Australia; CN:,China; SA: South Africa; UK, The United Kingdom; SA: South Africa; US: the United States.

### Levels of sodium content for processed meat and fish across the five countries
Table 1 shows the sodium content of processed meat and fish products across the five countries. Overall, for processed meat and fish products combined, China had the highest sodium level (1050 mg/100 g, IQR: 774–1473), ranking as the country with the saltiest products for both meat (1066 mg/100 g, IQR: 800–1450) and fish products (942 mg/100 g, IQR: 470–1867), followed by the USA, South Africa, Australia and the UK (432 mg/100 g, IQR: 236–786). Taking meat products alone, Australia had lower median sodium content (580 mg/100 g, IQR: 376–990) than the UK (590 mg/100 g, IQR: 275–904). Significant differences in sodium levels were seen in 18 subcategories among five countries. For example, the sodium content of roasted chicken in China was 4.5 times that of the UK (893 mg/100 g vs 197 mg/100 g) (p<0.001); chilled fish in China, 4.5 times that of the USA (1744 mg/100 g vs 389 mg/100 g) (p<0.001); pate and meat spreads in China, about 4 times that of Australia (1916 mg/100 g vs 480 mg/100 g) (p<0.001). However, the sodium content of bacon, frozen meat, salami and cured meats, dried meat and frozen fish in China was the lowest among five countries (figure 2). Taking bacon as an example, the median sodium contents ordered from highest to lowest were the USA (1667 mg/100 g), the UK (1612 mg/100 g), Australia (1150 mg/100 g), South Africa (1018 mg/100 g) and China (805 mg/100 g). Within each country, the sodium content also varied greatly across different subgroups with raw unflavoured meats being the lowest sodium content subcategory.

### Comparison of sodium content using Traffic Light criteria
Across the five countries, a large proportion of processed meat and fish products fell into the red and amber categories, with the highest proportion of green light products found in the UK, accounting for 12.66% of all meat and fish products. China had the largest proportion of red light (85.83%) and the smallest proportion of green light products (3.64%) (p<0.001). A similar difference was seen in processed meat products (p<0.001). For processed fish products, the highest proportion of green light products was observed in the USA (14.50%), followed by 12.84% in the UK. South Africa had the largest proportion of amber light products (84.73%) and the lowest proportion of red light (8.68%) and green light (6.59%) products among five countries (p<0.001) (figure 3).

### Comparison of sodium content to 2017 UK sodium reduction targets
In the 13 categories of processed meat and fish products, the countries with sodium contents reaching 2017 UK sodium reduction targets in descending order were the UK (26.6%), Australia (23.2%), South Africa (22.4%), the USA (18.4%) and China (7.1%). Statistically significant differences were observed among countries (p<0.001) for bacon, canned meat, frozen meat, meat burgers, sausage and hot dogs, other meat products and canned fish. The

**Table 1** Sodium content of processed meat and fish products across five countries (mg/100 g)

| | USA (n=17 098) | | | SA (n=946) | | | Australia (n=5673) | | | UK (n=885) | | | China (n=1898) | | | K-W H test |
|---|---|---|---|---|---|---|---|---|---|---|---|---|---|---|---|---|
| | n | Median | IQR | n | Median | IQR | n | Median | IQR | n | Median | IQR | n | Median | IQR | P Value |
| Total | 17098 | 655 | 353–981 | 946 | 571 | 362–876 | 5673 | 489 | 335–854 | 885 | 432 | 236–786 | 1898 | 1050 | 774–1473 | |
| Meat | 12954 | 768 | 474–1071 | 612 | 754 | 518–1020 | 3836 | 580 | 376–990 | 558 | 590 | 275–904 | 1641 | 1066 | 800–1450 | <0.001 |
| Meat alternative products | 372 | 478 | 386–607 | – | – | – | – | – | – | 10 | 413 | 236–550 | 1 | 1102 | 1102 | <0.001 |
| Bacon | 707 | 1667 | 1050–1857 | 36 | 1018 | 823–1155 | 289 | 1150 | 1020–1700 | 43 | 1612 | 1140–2162 | 33 | 805 | 750–1000 | 0.175 |
| Canned meat | 462 | 607 | 446–964 | 44 | 659 | 517–855 | 127 | 717 | 483–900 | 8 | 275 | 236–826 | 48 | 762 | 688–887 | <0.001 |
| Frozen meat | 1169 | 532 | 400–690 | 123 | 460 | 347–577 | 875 | 440 | 347–560 | 86 | 275 | 236–354 | 33 | 160 | 64–713 | 0.018 |
| Meat burgers | 824 | 476 | 305–647 | 47 | 638 | 500–794 | 162 | 475 | 390–584 | 6 | 393 | 315–472 | 7 | 612 | 486–703 | <0.001 |
| Salami and cured meats | 554 | 1607 | 1357–1750 | 25 | 1633 | 1415–1838 | 265 | 1410 | 1200–1600 | 5 | 1573 | 1376–1612 | 115 | 1200 | 949–1532 | <0.001 |
| Sausage and hot dogs | 2817 | 829 | 696–974 | 133 | 814 | 684–935 | 456 | 704 | 565–900 | 36 | 550 | 452–747 | 269 | 991 | 832–1111 | <0.001 |
| Sliced meat | 1937 | 875 | 750–1088 | 84 | 900 | 745–1100 | 359 | 989 | 816–1100 | 173 | 668 | 590–865 | 23 | 1132 | 845–1250 | <0.001 |
| Dried meat | 1383 | 1536 | 1036–1929 | 43 | 2144 | 1682–2280 | 126 | 1760 | 1400–2000 | – | – | – | 281 | 1509 | 1153–1760 | <0.001 |
| Pate and meat spreads | 83 | 679 | 518–911 | 14 | 789 | 438–861 | 89 | 480 | 310–603 | 28 | 629 | 550–708 | 5 | 1916 | 1670–2490 | <0.001 |
| Kebabs | 2 | 493 | 462–525 | – | – | – | 38 | 408 | 294–504 | 1 | 393 | 393 | – | – | – | <0.001 |
| Other meat products | 427 | 589 | 94–1071 | 26 | 865 | 560–1070 | 83 | 570 | 340–925 | 13 | 275 | 236–472 | 649 | 1050 | 782–1390 | – |
| Raw flavoured meats | 501 | 446 | 254–750 | 24 | 465 | 356–580 | 678 | 368 | 245–502 | 27 | 197 | 197–315 | 42 | 563 | 382–763 | <0.001 |
| Whole hams and similar products | 4 | 848 | 750–1518 | 2 | 839 | 744–934 | 80 | 1080 | 1000–1400 | 12 | 1081 | 983–1356 | 103 | 1039 | 940–1250 | <0.001 |
| Roasted chicken | 2 | 563 | 357–768 | 1 | 415 | 415 | 37 | 359 | 271–548 | 3 | 197 | 118–236 | 9 | 893 | 693–996 | 0.977 |
| Raw unflavoured meats | 1710 | 71 | 63–402 | 10 | 69 | 58–120 | 172 | 66 | 50–323 | 62 | 79 | 79 | 23 | 122 | 70–234 | <0.001 |
| Fish | 4144 | 364 | 208–529 | 334 | 356 | 265–453 | 1837 | 395 | 286–540 | 327 | 354 | 236–550 | 257 | 942 | 470–1867 | 0.011 |
| Canned fish | 1219 | 388 | 299–467 | 168 | 353 | 280–400 | 821 | 380 | 309–472 | 66 | 354 | 315–393 | 138 | 902 | 599–1586 | <0.001 |
| Chilled fish | 147 | 389 | 171–691 | 36 | 449 | 226–798 | 332 | 587 | 324–917 | 126 | 511 | 157–747 | 53 | 1744 | 370–5072 | <0.001 |
| Frozen fish | 2733 | 347 | 152–541 | 100 | 295 | 169–434 | 559 | 340 | 225–449 | 117 | 275 | 197–354 | 36 | 131 | 73–715 | <0.001 |
| Other fish | 45 | 5389 | 3813–6000 | 30 | 451 | 362–580 | 125 | 860 | 426–4990 | 18 | 550 | 432–747 | 30 | 1305 | 1147–1644 | <0.001 |

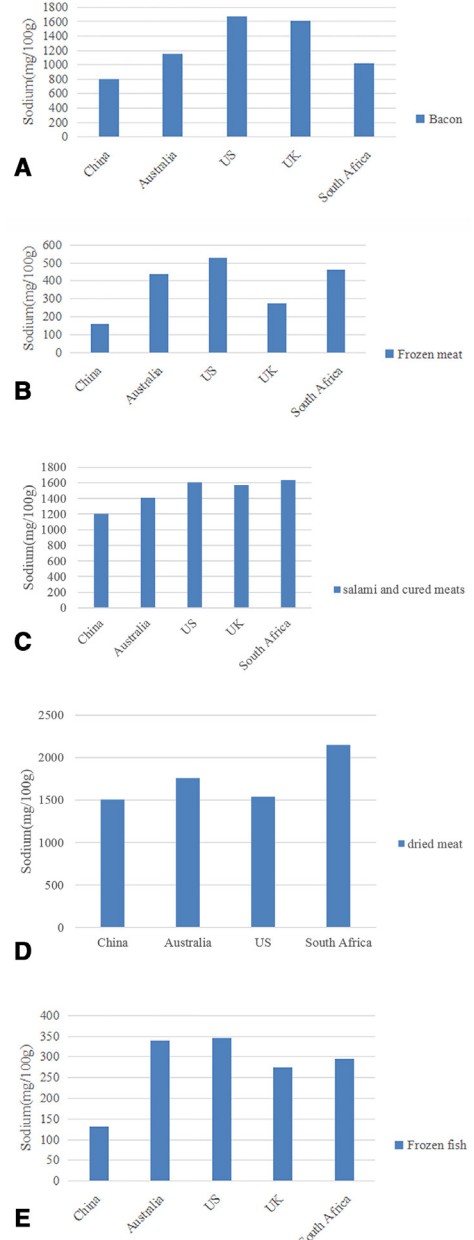

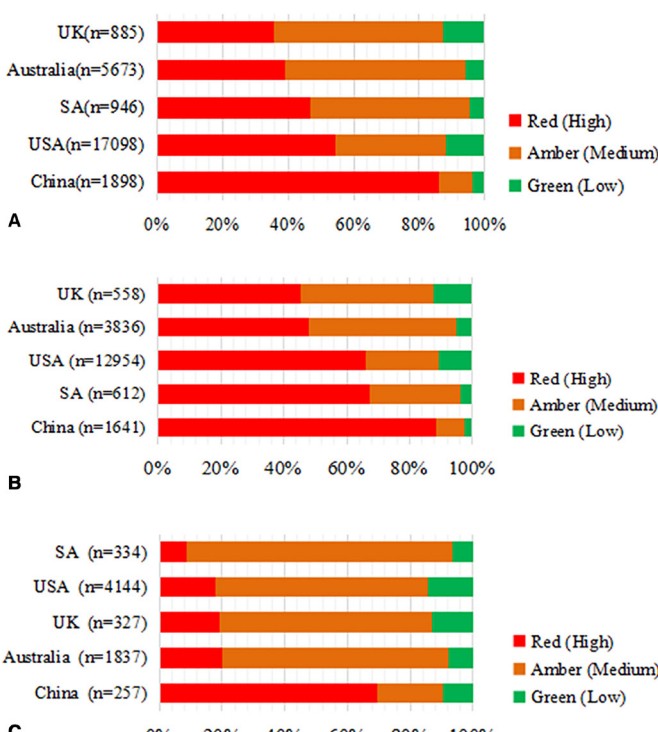

**Figure 3** (A) Sodium content Traffic Light on processed meat and fish products among five countries. (B) sodium content Traffic Light on processed meat products among five countries. (C) Sodium content Traffic Light on processed fish products among five countries. Red, high; amber, medium; green, Low.

**Figure 2** (A) Sodium content of bacon among five countries: bacon; (B) Sodium content of frozen meat among five countries: frozen meat; (C) sodium content of salami and cured meats among five countries: salami and cured meats; (D) sodium content of dried meat among five countries: dried meat; (E) sodium content of frozen fish among five contries: frozen fish.

UK had the highest percentage of products achieving the targets except for bacon products, in which only 14.0% of bacon products in the UK reached the target, lower than that in the USA (28.2%), Australia (50.2%), SA (75.0%) and China (84.9%) (table 2).

### Contribution of sodium content per 100 g to WHO daily sodium intake recommendation

Table 3 shows the sodium intake contribution from the consumption of processed meat and fish products. If 100 g meat and fish products was consumed, the sodium

intake would account for 47.2% of the WHO recommended maximum daily intake (2000 mg/day) on average in China, followed by the USA (47.1%), South Africa (36.9%), Australia (34.6%) and the UK (27.1%). Each country had its own major sodium contributors. For example, the sodium contribution values were the highest for pate and meat spreads (95.8%) and chilled fish (87.2%) in China, but very low in the other four countries. Several food categories had relative high sodium intake contribution, highlighted with red or yellow across the five countries. They were dried meat, salami and cured meats, bacon, sliced meat, and whole hams and similar products.

### DISCUSSION

This study provides the first detailed comparison of sodium content in processed meat and fish products among five countries. The results show extremely wide discrepancy within and between countries. Overall, processed meat and fish products in the UK had the lowest median sodium content, and China had the highest sodium in both meat and fish products. The sodium content of meat and fish products in each country was high compared with 'Traffic Light' criteria with only 10% of the products in the UK and USA and no more than 5% in China and South Africa falling into the green light group. The percentage of products meeting 2017 UK sodium reduction targets

**Table 2** Number and percentage of products with sodium content meeting the 2017 UK sodium targets

| Product categories | 2017 UK sodium targets (mg/100 g)* | China n (%) | UK n (%) | Australia n (%) | SA n (%) | USA n (%) | P value |
|---|---|---|---|---|---|---|---|
| Total | – | 101 (7.1) | 135 (26.6) | 833 (23.2) | 154 (22.4) | 1927 (18.4) | <0.001 |
| Meat alternative products | 500 | 0 (0.0) | 5 (50.0) | – | – | 212 (57.0) | 0.907 |
| Bacon† | 1152 | 28 (84.9) | 6 (14.0) | 145 (50.2) | 27 (75.0) | 199 (28.2) | <0.001 |
| Canned meat | 272 | 0 (0.0) | 4 (50.0) | 5 (3.9) | 0 (0.0) | 24 (5.2) | <0.001 |
| Frozen meat† | 272 | 17 (51.5) | 36 (41.9) | 87 (9.9) | 18 (14.6) | 139 (11.9) | <0.001 |
| Meat burgers | 352 | 1 (14.3) | 2 (33.3) | 27 (16.7) | 4 (8.5) | 249 (30.2) | <0.001 |
| Salami and cured meats† | 652 | 5 (4.4) | 0 (0.0) | 2 (0.8) | 0 (0.0) | 18 (3.3) | 0.08 |
| Sausage and hot dogs | 600 | 7 (2.6) | 21 (58.3) | 147 (32.2) | 14 (10.5) | 350 (12.4) | <0.001 |
| Sliced meat | 272 | 1 (4.4) | 5 (2.9) | 9 (2.5) | 0 (0.0) | 32 (1.7) | 0.218 |
| Kebabs | 352 | – | 0 (0.0) | 14 (36.8) | – | 0 (0.0) | – |
| Other meat products | 300 | 29 (4.5) | 8 (61.5) | 17 (20.5) | 2 (7.7) | 150 (35.1) | <0.001 |
| Whole hams and similar products† | 652 | 0 (0.0) | 1 (8.3) | 3 (3.8) | 0 (0.0) | 0 (0.0) | 0.036 |
| Roasted chicken | 272 | 0 (0.0) | 3 (100.0) | 10 (27.0) | 0 (0.0) | 0 (0.0) | 0.189 |
| Canned fish† | 360 | 13 (9.4) | 44 (66.7) | 367 (44.7) | 89 (53.0) | 554 (45.5) | <0.001 |

*The maximum sodium targets of each category were selected for ease of comparison, and the average targets were used where maximum target was not provided.
†Average sodium targets.

were generally low ranging from 7.1% (China) to 26.6% (the UK). A 100 g serving size of processed meat and fish products could on average contribute to one half/third of WHO daily maximum sodium intakes in all countries.

The amount of sodium intake from prepackaged food differs in different countries. In developed countries like Australia, USA and the UK, processed foods provide 75%~80% of sodium intake.[20][21] It was reported that processed meat products accounted for about 20% of daily meat consumption and contributed to around 10% daily sodium intake in Australia.[11] In South Africa, processed meat was also a major sodium source other than bread among processed foods, which contributed to about 50% of sodium intake.[21] In China, however, 70%~80% of sodium came from home cooking with a remarkable increase from consumption of processed foods and meals out of home in recent years. Sodium intake from packaged meat and fish products is an emerging concern.[22]

One strategy to reduce sodium intake from packaged products is to encourage consumers to replace high sodium products with low sodium products. For example, choosing raw unflavoured meats instead of salami and cured meats would decrease the sodium intake from these foods by 10-fold to 20-fold in all five countries. However, different subcategories of meat and fish products have distinct different organoleptic properties, which coupled with the convenience of pre-prepared products is the main driver for consumers' choice.[23] Therefore, development of new products with the same or better flavour and less sodium should be encouraged. In addition, Front-of-Pack labelling such as Traffic Light and Health Star Rating labelling as well as consumer awareness campaigns

may increase consumer acceptability and demand for healthier products.[21][24]

It is not easy to simply replace or reformulate the high sodium products that already exists for years. However, the large difference in sodium content of similar products in different countries, and the difference in sodium content among different brands within the same countries, indicate that there is still a lot of room for salt reduction. Product features regarding satisfying flavour, texture, safety and stability have been the key considerations for manufacturers, but attention should also be paid for three situations. First, product formulations might have been lagged behind consumers' requirement for less sodium products. Second, many manufacturers may resist reformulation due to unfounded concern for flavour acceptance and safety.[25] Third, a 10%–15% reduction in sodium will go undetected, and the product reformulation could be done step by step.[8][25]

Setting sodium targets for processed foods is an effective way to reduce sodium contents of packaged foods.[4][11][13][24] In the five countries, China had the saltiest meat and fish products among the countries, which is likely due to the lack of sodium targets to limit the sodium added to the products. The remaining four countries have set voluntary or mandatory sodium targets for meat and fish products along with comprehensive sodium reduction policies/programs. The UK has issued four sets of voluntary sodium targets for over 80 categories of processed foods since 2006 and has set up a successful sodium reduction model for other countries through this incremental sodium reduction strategy.[7] Following the UK, the USA and Australia set the voluntary sodium targets

**Table 3** Sodium intake contribution values (%) of processed meat and fish products*

| Food categories | China | UK | Australia | SA | USA |
|---|---|---|---|---|---|
| All categories | 47.2 | 27.1 | 34.6 | 36.9 | 47.1 |
| Meat alternative products | 55.1 | 20.7 | – | – | 23.9 |
| Bacon | 40.3 | 80.6 | 57.5 | 50.9 | 83.4 |
| Canned meat | 38.1 | 13.8 | 35.9 | 33.0 | 30.4 |
| Frozen meat | 8.0 | 13.8 | 22.0 | 23.0 | 26.6 |
| Meat burgers | 30.6 | 19.7 | 23.8 | 31.9 | 23.8 |
| Salami and cured meats | 60.0 | 78.7 | 70.5 | 81.7 | 80.4 |
| Sausage and hot dogs | 49.6 | 27.5 | 35.2 | 40.7 | 41.5 |
| Sliced meat | 56.6 | 33.4 | 49.5 | 45.0 | 43.8 |
| Dried meat | 75.5 | – | 88.0 | 107.2 | 76.8 |
| Pate and meat spreads | 95.8 | 31.5 | 24.0 | 39.5 | 34.0 |
| Kebabs | – | 19.7 | 20.4 | – | 24.7 |
| Other meat products | 52.5 | 13.8 | 28.5 | 43.3 | 29.5 |
| Raw flavoured meats | 28.2 | 9.9 | 18.4 | 23.3 | 22.3 |
| Whole hams and similar products | 52.0 | 54.1 | 54.0 | 42.0 | 42.4 |
| Roasted chicken | 44.7 | 9.9 | 18.0 | 20.8 | 28.2 |
| Raw unflavoured meats | 6.1 | 4.0 | 3.3 | 3.5 | 3.6 |
| Canned fish | 45.1 | 17.7 | 19.0 | 17.7 | 19.4 |
| Chilled fish | 87.2 | 25.6 | 29.4 | 22.5 | 19.5 |
| Frozen fish | 6.6 | 13.8 | 17.0 | 14.8 | 17.4 |
| Other fish | 65.3 | 27.5 | 43.0 | 22.6 | 269.5 |

*The contribution value, calculated as median sodium content (mg/100 g)/2000 (mg/day)*100%, was a ratio of daily sodium intake from 100 g product against the WHO maximum sodium recommendation (2000 mg/day), assuming the daily consumption of processed meat and fish food for a person were 100 g/day. The contribution values were highlighted as red, yellow and green to represent high (>66%), medium (>33%, ≤66%) and low (≤33%) contribution to sodium intake, respectively.

for various processed foods through the National Salt Reduction Initiative in 2008 in the USA and the Food and Health Dialogue in 2010 in Australia, respectively.[20 24] South Africa became the first country to regulate legislated sodium limits for a range of food products in 2012.[12] The results of comparing sodium contents against the latest 2017 UK sodium reduction targets showed that the UK had the highest proportion of products achieving the targets, followed by Australia, South Africa, USA and China. This, to some extent, might be relevant to the implementation of the incremental sodium reduction strategies.

Target implementation is also critical. Our results showed that the proportion of meat and fish products that met the sodium reduction targets was low across all the countries. Even for the best, the UK, the target-achieving rate was only 26.6% for all meat and fish products, which was much lower than the target-achieving rate for noodles (90%) and sauces (70%).[4 13] Some subcategories of meat products such as bacon even had the highest sodium content in the UK among the countries. These suggest that robust implementation and monitoring of the voluntary targets are needed. The 2017 UK sodium reduction targets were more rigorous compared with that

of other countries. Studies have shown that in Australia, South Africa and the USA, about half of meat products met their own national targets.[11 12 20] In summary, the sodium-lowering targets provide a level playing field within a country. Many food manufacturers are trying to work towards the targets. This finding also indicates that technical issues should not be a barrier for manufacturers to reformulate their foods.

With development and urbanisation, more and more countries have realised the increasing challenge of prepackaged food to health. Although not surprising to many people, the specific findings in this study could be a good reference in developing specific strategies to promote sodium reduction. To achieve this, several questions could be considered. What the gap and space is for a country in sodium reduction for prepackaged food? Which products should be targeted on first? Whether and how to adopt the target setting strategy, mandatory or voluntary? And how to overcome the barriers from manufacturers who may be reluctant to reformulate their product by arguing that salt reduction would shorten the shelf life?

The key strength of this study is that it is the first cross-sectional survey of the sodium content of processed meat

and fish products in five countries. The standardised methods for data collection and processing, including standardised food categorisation, have ensured the comparability of the data. There are several potential limitations of this study. First, products were obtained only in selected stores at a specific time in each country, and the selected stores were major supermarket chains with a large market share but could not represent all stores within the countries. Second, we did not capture food-purchasing data to quantify actual sodium consumption of processed meat and fish products, although the crowd-sourcing element of the data collection may somehow reflect what consumers have eaten. Future studies should consider using more reliable product sales data or consumption data to estimate the actual sodium intake from processed meat and fish products in each country. Third, the duration of data collection varies from 2015 to 2018 in China, 2013 to 2017 in the UK, 2014 to 2017 in Australia, 2015 to 2017 in South Africa and 2012 to 2017 in the USA. During these periods, although very slow, product reformulation may have occurred due to growing global interest in sodium reduction. To make full use of the data and due to the lack of track records for each product, we did not compare the five countries over the same time and were not able to identify and exclude the outdated products during analysis.

## CONCLUSION

The sodium content of meat and fish products in all the selected countries was very high with a 100 g serving size of meat and fish products contributing to one half/ third of WHO recommended maximum daily sodium intake. There are large differences in sodium levels of packaged foods among the five countries with different sodium reduction policies. This implies that the target-based strategy is effective in lowering sodium levels in foods. Therefore, setting feasible or further lower sodium targets is urgent. Regular evaluation is also needed to ensure its robust implementation.

**Author affiliations**
[1]Department of nutrition and food hygiene, Hebei Medical University, Shijiazhuang, China
[2]Peking University Health Science Centre, The George Institute for Global Health, Beijing, China
[3]Faculty of Medicine, University of New South Wales, Australia, Sydney, New South Wales, Australia
[4]University of Hong Kong, Hong Kong, China
[5]Food Policy, The George Institute for Global Heath, Sydney, New South Wales, Australia
[6]Centre for Environmental and Preventive Medicine, Wolfson Institute of Preventive Medicine, Barts and The London School of Medicine & Dentistry, Queen Mary University of London, London, UK
[7]Wolfson Institute of Preventive Medicine, Queen Mary University of London, London, UK
[8]Discovery Vitality, Johannesburg, South Africa, South African, South Africa
[9]School of Computing, Beihang University, Beijing, China
[10]Chinese Center for Disease Control and Prevention National Institute for Nutrition and Health, Beijing, China
[11]Diabetes Program, The George Institute at Peking University Health Science Center, Beijing, China

**Acknowledgements** We thank Prof Feng J He, Monique Tan, Prof Graham MacGregor from Queen Mary University of London provided data of the UK. Prof Bruce Neal from The George Institute for Global Health permitted and coordinated the data sharing. Blaire Jesseph from Label Insight in the USA and Terry Harris from Vitality South Africa collaborated with The George Institute in developing the dataset of the USA and South Africa, respectively. Fraser Taylor, Kylie Howes and Lizzy Dunford from The George Institute FoodSwitch team mapped the data and directly provided data of Australia, South Africa and USA. Yu Liu from Beihang University provided data of China. Huijun Wang from National Institute for Nutrition and Health at Chinese Centre for Disease Control and Prevention provided data of China. We appreciate Jimmy Louie from The University of Hong Kong for his contribution to the revision of the very first draft. And thank National Financial Projects for its financial support.

**Contributors** PZ, YM, YLi and CG designed the study. LH, MT, FJH, TH, GAM, YLiu and HW provided the data. YS, YW, JD and LD analysed the data. YS prepared the first draft of the manuscript. YLi, CG, PZ and YM made critical revisions to the manuscript. All authors reviewed and approved the final draft.

**Funding** This research was funded by National Key R&D Program of China, grant number 2016YFC1300200, National Financial Projects (131031107000160004) and the National Institute for Health Research (NIHR, NIHR Global Health Research Unit Action on Salt China at Queen Mary University of London) using Official Development Assistance (ODA) funding (16/136/77).

**Competing interests** None declared.

**Patient and public involvement** Patients and/or the public were not involved in the design, or conduct, or reporting, or dissemination plans of this research.

**Patient consent for publication** Not required.

**Provenance and peer review** Not commissioned; externally peer reviewed.

**Data availability statement** Data are available upon reasonable request. Nutrition information of 19601 meat and 6899 fish products was collected using the FoodSwitch mobile application from China, the United Kingdom (UK), Australia, South Africa and the United States (US) from 2012 to 2018.The data of China can be linked yli@georgeinstitute.org.cn and zpuhong@georgeinstitute.org.cn.The data of UK can be linked m.tan@qmul.ac.uk; f.he@qmul.ac.uk; g.macgregor@qmul.ac.uk.The data of Australia can be linked ftaylor@georgeinstitute.org.au.The data of South Africa can be linked TerryH@discovery.co.za.The data of US can be linked bjesseph@labelinsight.com. These data are used with the permission of the authors.

**ORCID iDs**
Yuzhu Song http://orcid.org/0000-0003-3623-3052
Monique Tan http://orcid.org/0000-0003-4287-5553

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
