## [Reviewer comments · BMJ Open]

ARTICLE DETAILS

TITLE (PROVISIONAL)	Cross-sectional comparisons of sodium content in processed meat and fish products among five countries - potential for feasible targets and reformulation
AUTHORS	Song, Yuzhu; Li, Yuan; Guo, Chunlei; Wang, Yishan; Huang, Liping; Tan, Monique; He, Feng; Harris, Terry; MacGregor, Graham; Ding, Jingmin; Dong, Le; Liu, Yu; Wang, Huijun; Zhang, Puhong; Ma, Yuxia

VERSION 1 – REVIEW

REVIEWER	Charlton, Karen University of Wollongong, School of Health Sciences
REVIEW RETURNED	21-Dec-2020

GENERAL COMMENTS	This study reports on a five country comparison of the sodium content of processed meat and fish products using food label information collecting using the Foodswitch app. The data is clearly presented and the manuscript written succinctly. However, there are many grammatical errors throughout, as listed below. The entire manuscript should be thoroughly checked for English language prior to its resubmission. The study findings have important public health implications in terms of identifying potential for reformulation of these products by food producers, in order to remove salt from the food supply. This is particularly relevant for countries that still have very high population level salt intakes, such as China. I therefore recommend the paper be published in BMJ Open, pending suggested minor changes are made. Introduction Well written introduction that provides rationale why processed meat and fish products are the target of the current between-country comparison. For example, these foods alone provide over half of the recommended daily sodium intake in China. Page 6, line 3: Typo - In developed countries where more than three quarters of sodium coming (replace with “comes”) from processed foods... Line 26: “This target-based approach has been shown to be effective in reducing sodium content in many food products^{11 12} and, for the same food category, the sodium level is much higher in the countries without sodium reduction target than those with the target.⁴” Replace “target” with “targets”
---

Line 35: "The median sodium contents were on average 4.4-fold greater in Chinese sauces compared with their UK equivalents.13"
Replace "contents" with "content".

Methods

Page 7, line 20: "...professionally trained data entry clerks then entered the information displayed in products package"

Replace "package" with "packaging"

Data collection occurred between 2012 and 2018 - this is a long period during which product reformulation may have occurred as a result of growing interest, globally, in policies to reduce salt (ie. voluntary and mandatory targets for processed foods). This issue needs to be addressed in the Discussion.

Page 7, Line 37: "We also obtained processed meat and fish products data from the US, which were shared by Label Insight Inc. to The George Institute for non-profit research."

Change to "We also obtained data on processed meat and fish products from the US from Label Insight Inc. that was shared with the George Institute for non-profit research."

Page 7, line48: "Processed meat products were further classified to the following 16 subcategories: meat-free products..."

How can meat-free products be classified as processed meat products? This does not make sense to me.

Page 8, line 12: "For identical products with same sodium content in different package sizes, it is regarded as a duplicate product, only one product was included."

Change to "In the case of identical products with the same sodium content, but available in different package sizes, these were regarded as duplicates and only one product was included."

Page 9, first para: It is not clear why data is presented as a serving size rather than merely per 100g. A serving size was estimated as 94g for processed meat and rounded to 100g, so it is essentially the same thing. It cannot be assumed that serving sizes for these foods is the same across countries therefore I suggest changing the description to "per 100g".

Page 9, first para: "The percentage contribution of sodium intake from each serving of meat or fish products towards the recommended daily sodium intake was coloured into red, yellow, and green respectively to represent if the percentage is in the upper (>66%), middle (>33%, ≤66%) and lower (≤33%) range. "

Change to "The percentage contribution of sodium intake per 100g serving of meat or fish products towards the recommended daily sodium intake was graphically presented in coloured bar charts, with red, yellow, and green representing upper (>66%), middle (>33%, ≤66%) or lower (≤33%) range, respectively.

Page 11, line 6: Incorrect use of tense. "Table1 showed the sodium .." Change to "shows"

Results

Page 11, line 50: "Across the five countries, a large part of processed meat and fish products fell into the red and amber category, the highest proportion of green light was in the UK, accounting for 12.66% of the meat and fish products. China had

the largest proportion of red light (85.83%) and the smallest proportion of green light (3.64%) ($\chi^2=1101.13$,
Change to: “Across the five countries, a large proportion of processed meat and fish products fell into the red and amber categories, with the highest proportion of green light products found in the UK, accounting for 12.66% of all meat and fish products. China had the largest proportion of red light (85.83%) and the smallest proportion of green light (3.64%) ($\chi^2=1101.13, P<0.001$) products.

Page 12, line 3: “A similar differences...” should be singular, i.e. “difference”

Page 12, line 5: “For processed fish products, the highest green light was 14.50% in the US, followed by 12.84% in the UK, and South Africa had the largest portion of amber light (84.73%) and the lowest red light (8.68%) and green light (6.59%) ($\chi^2=277.49$, $p<0.001$).(Figure 2-1,2-2,2-3)

Change to:” “For processed fish products, the highest proportion of green light products was observed in the US (14.50%), followed by 12.84% in the UK. South Africa had the largest proportion of amber light (84.73%) and the lowest proportion of red light (8.68%) and green light (6.59%) products of all five countries ($\chi^2=277.49$, $p<0.001$).(Figure 2-1,2-2,2-3)

Page 12, line 23: “...from high to low...”

Change to “...in descending order”

Page 12, line 32: “The UK had the highest percentage of achieving the targets except in the bacon category, only..”

Change to “...highest percentage of products achieving..”

Page 14, first line: “7.1% in the US, 36.9% in South Africa, 34.6% in Australia and 27.1% in the UK”

It is unclear what these figures refer to, please clarify. Does not seem to be average for products, as shown in Table 3.

Discussion

Page 15, first line: Change tense from “provided” to “provides”. Same for line 9, change “showed” to “shows”.

Line 27: “averagely” is not a word, change to “on average”

Page 16, first line: “...unflavoured meats instead of salami and cured meats would decrease ten to twenty times of sodium intake in all countries.”

Change to “...unflavoured meats instead of salami and cured meats would decrease the sodium intake from these foods by ten to twenty fold in all five countries.”

Page 16, line 17: Replace “big” with “large”

Line 42: Incorrect tense “...the UK had issued...” Change “had” to “has”

Page 17, lines 1-10: “targets were consistent with the median meat and fish products showing the highest proportion of meeting the targets in the UK, followed by Australia, South Africa, US and China, which to a certain extent reflected the implementation of sodium reduction policy.”

	This statement is incorrect - the paper does not evaluate whether sodium reduction policies have been implemented. Please reword. Line 31: "Other studies comparing the sodium contents against the country-specific sodium showed higher proportion with about half of meat products meeting the individual targets in Australia, South Africa and the US.11 12 20. It was therefore worth learning for China to take into account both the technical feasibility and consumer acceptability if sodium targets were to be set in the future." Please rewrite these sentences, incorrect English language. Page 18, line 13: "Second, we did not capture household consumer panel food-purchasing data to quantify actual sodium consumption of processed meat and fish products." The term "household consumer panel food-purchasing data" is unfamiliar to me. Page 18, line 20: "using proper product sales data". Replace word "proper" with "more reliable", or something similar. Conclusions Page 18, Line 33, replace word "big" with "large" There were big variations within and between the five countries with different sodium reduction policies, which implies great potential of sodium reduction in meat and fish products by setting feasible sodium reduction targets in countries without sodium reduction program and sustaining robust implementation and monitoring of the targets in the countries with sodium targets in place, as well as selection of less salted food by consumers. Please rewrite this sentence to be more grammatically correct.
--	---

REVIEWER	Zand, Nazanin University of Greenwich
REVIEW RETURNED	17-Jan-2021

GENERAL COMMENTS	The discussion is very limited. The world Reformulation is misleading in the title as it is not discussed or addressed by the study at all. This study would be enriched if the results were correlated with the prevalence of the disease Hypertension in each country and then compared. Also a major limitation is the missing insight into consumer purchase trend and buying behaviours. Perhaps these data could have been obtained from the sales or market research? As this study stands, it is not clear Which new knowledge is adding to the scientific community? the issue of salt in processed food as mentioned in the discussion, is already proved and is an area that is well explored. English can benefit from proof reading to make the text more concise. Attached pdf in adobe provides further comments. You need to open with adobe to see the comments. The reviewer provided a marked copy with additional comments. Please contact the publisher for full details.
--

VERSION 1 – AUTHOR RESPONSE

Reviewer 1-grammatical problems

General response: Thank you for correcting the grammatical problems of this paper. Here are our corrections, which are highlighted in the article.

1. Page 6, line 3: Typo - In developed countries where more than three quarters of sodium coming (replaced with "intake comes") from processed foods...

Response: We have replaced "coming" with "intake comes"

2. Line 26: "This target-based approach has been shown to be effective in reducing sodium content in many food products^{11 12} and, for the same food category, the sodium level is much higher in the countries without sodium reduction target than those with the target.⁴"

Response: We have replaced the sentence with "This targets-based approach has been shown to be effective in reducing sodium content for many food products.^{11 12} Within the same food category, the sodium level is much lower in food products in countries with sodium reduction targets than those without the targets.⁴"

3. Line 35: "The median sodium contents were on average 4.4-fold greater in Chinese sauces compared with their UK equivalents.¹³"

Response: We have replaced the sentence with "A study comparing the sodium content of sauce products between UK and China showed that the median sodium content of sauce products was 4.4-fold less in UK when compared with the Chinese equivalents.¹³".

4. Methods

Page 7, line 20: "...professionally trained data entry clerks then entered the information displayed in products package"

Response: We have replace the sentence with "The information displayed on the packages, including product, nutrition and ingredient information, was then entered into a uniform web-based data management system by professionally trained clerks."

5. Page 7, Line 37: "We also obtained processed meat and fish products data from the US, which were shared by Label Insight Inc. to The George Institute for non-profit research."

Response: We have rewritten the sentence as "We also obtained data on processed meat and fish products from the US through Label Insight Inc. for non-profit research."

6. Page 8, line 12: "For identical products with same sodium content in different package sizes, it is regarded as a duplicate product, only one product was included."

Response: We have rewritten the sentence as "In the case of identical products with the same sodium content, but available in different package sizes, these were regarded as duplicates and only one product was included."

7. Page 9, first para: It is not clear why data is presented as a serving size rather than merely per 100g. A serving size was estimated as 94g for processed meat and rounded to 100g, so it is essentially the same thing. It cannot be assumed that serving sizes for these foods is the same across countries therefore I suggest changing the description to "per 100g".

Response: We have replaced "each serving" with "per 100g" and rewritten the relevant parts in Methods and Results sections. This makes the description simpler. Many thanks.

8. Page 9, first para: "The percentage contribution of sodium intake from each serving of meat or fish products towards the recommended daily sodium intake was coloured into red, yellow, and green respectively to represent if the percentage is in the upper (>66%), middle (>33%, ≤66%) and lower (≤33%) range."

Response: To make the calculation of contribution values much clearer, we have rewritten this part as

“To measure the sodium burden caused by consumption of processed meat and fish products, a sodium intake contribution value was calculated for each category of food products. It was a ratio of daily sodium intake from 100g product against the WHO maximum sodium recommendation (2000 mg/d), assuming the daily consumption of processed meat and fish food for a person were 100 g per day. For each category of the food products, the contribution value was calculated as median sodium content (mg/100g) / 2000 (mg/d) * 100%, and was highlighted as red, yellow and green respectively to represent high (>66%), medium (>33%, ≤66%) and low (≤33%) sodium intake contribution.”

9. Page 11, line 6: Incorrect use of tense. “Table1 showed the sodium ..”

Response: We have changed “showed” to “shows”

10. Results

Page 11, line 50: “Across the five countries, a large part of processed meat and fish products fell into the red and amber category, the highest proportion of green light was in the UK, accounting for 12.66% of the meat and fish products. China had the largest proportion of red light (85.83%) and the smallest proportion of green light (3.64%) ((c2=1101.13, p<0.001)

Response: We have changed the sentence to: “Across the five countries, a large proportion of processed meat and fish products fell into the red and amber categories, with the highest proportion of green light products found in the UK, accounting for 12.66% of all meat and fish products. China had the largest proportion of red light (85.83%) and the smallest proportion of green light products (3.64%) (p<0.001).

11. Page 12, line 3: “A similar differences...”

Response: We have changed “differences” to “difference”

12. Page 12, line 5: “For processed fish products, the highest green light was 14.50% in the US, followed by 12.84% in the UK, and South Africa had the largest portion of amber light (84.73%) and the lowest red light (8.68%) and green light (6.59%)(c2=277.49, p<0.001).(Figure 2-1, 2-2, 2-3)

Response: We have changed the sentence to “For processed fish products, the highest proportion of green light products was observed in the US (14.50%), followed by 12.84% in the UK. South Africa had the largest proportion of amber light products (84.73%) and the lowest proportion of red light (8.68%) and green light (6.59%) products among the five countries (p<0.001).(Figure 3-1, 3-2, 3-3)”

13. Page 12, line 23: “...from high to low...”

Response: Thanks, we have changed it to “...in descending order”

14. Page 12, line 32: “The UK had the highest percentage of achieving the targets except in the bacon category, only..”

Response: We have changed the sentence to “The UK had the highest percentage of products achieving the targets except for bacon products, in which only 14.0% of bacon products in the UK reached the target, lower than that in the US (28.2%), Australia (50.2%), SA (75.0%) and China (84.9%). ”

15. Discussion

Page 15, first line: Change tense from “provided” to “provides”. Same for line 9, change “showed” to “shows”.

Response: Changed as suggested.

16. Line 27: “averagely” is not a word, change to “on average”

Response: Changed as suggested.

17. Page 16, first line: “...unflavoured meats instead of salami and cured meats would decrease ten to

twenty times of sodium intake in all countries.”

Change to “...unflavoured meats instead of salami and cured meats would decrease the sodium intake from these foods by ten to twenty fold in all five countries.”

Response: Changed as suggested.

18. Page 16, line 17: delete “big”

Response: Changed as suggested

19. Line 42: Incorrect tense “...the UK had issued...” Change “had” to “has”

Response: Changed.

20. Page 18, line 13: “Second, we did not capture household consumer panel food-purchasing data to quantify actual sodium consumption of processed meat and fish products.” Change to: “Second, we did not capture food-purchasing data to quantify actual sodium consumption of processed meat and fish products.”

Response: have accepted, many thanks.

21. Page 18, line 20: “using proper product sales data”. Replace word “proper” with “more reliable”, or something similar.

Response: We have changed “proper” to “more reliable”.

22. Conclusions

Page 18, Line 33, replace word “big” with “large”

Response: Done.

Reviewer 1-Question answer:

23. Data collection occurred between 2012 and 2018 - this is a long period during which product reformulation may have occurred as a result of growing interest, globally, in policies to reduce salt (ie. voluntary and mandatory targets for processed foods). This issue needs to be addressed in the Discussion.

Response: Thank you for this valuable feedback. We added this as one of the limitations in Discussion section. “Thirdly, the duration of data collection varies from 2015 to 2018 in China, 2013 to 2017 in the UK, 2014 to 2017 in Australia, 2015 to 2017 in South Africa and 2012 to 2017 in the US. During these periods, although very slow, product reformulation may have occurred due to growing global interest in sodium reduction. To make full use of the data and due to the lack of track records for each product, we did not compare the 5 countries over the same time and were not able to identify and exclude the outdated products during analysis.”

24. Page 7, line48: “Processed meat products were further classified to the following 16 subcategories: meat-free products...”

How can meat-free products be classified as processed meat products? This does not make sense to me.

Response: Thanks for pointing out this. We have included the meat-free products in meat products for several reasons. First, all the five countries adopt the same food categorization system in which meat-free products are categorized into meat products. Second, although not real meat products, meat-free products have been produced to replace certain meat products with similar flavor and nutrition. Third, treated as meat replacement, meat-free products usually adopt similar processing technology and have similar sodium content to real meat products. In order to make the name more appropriate, and also suggested by Reviewer 2, we changed the name of this category to “meat alternative products”.

25. Page 14, first line: “47.1% in the US, 36.9% in South Africa, 34.6% in Australia and 27.1% in the UK” It is unclear what these figures refer to, please clarify. Does not seem to be average for products, as shown in Table 3.

Response: These figures are sodium intake contribution values which are a ratio of daily sodium intake from 100g product against the WHO maximum sodium recommendation (2000 mg/d), assuming the consumption of processed meat and fish food products for a person is 100g per day. For each category of the food products, the contribution value was calculated as median sodium content (mg/100g) / 2000 (mg/d) * 100%, and was highlighted as red, yellow and green respectively to represent high (>66%), medium (>33%, ≤66%) and low (≤33%) sodium intake contribution in Table 3. Corresponding changes have been made in Methods, Results, Table 3, the note to Table 3, and Discussion sections.

26. Page 17, lines 1-10: “targets were consistent with the median meat and fish products showing the highest proportion of meeting the targets in the UK, followed by Australia, South Africa, US and China, which to a certain extent reflected the implementation of sodium reduction policy.”

This statement is incorrect - the paper does not evaluate whether sodium reduction policies have been implemented. Please reword.

Response: Thank you for your comments. This sentence has been modified to “The results of comparing sodium contents against the latest 2017 UK sodium reduction targets showed that the UK had the highest proportion of products achieving the targets, followed by Australia, South Africa, US and China. This, to some extent, might be relevant to the implementation of the incremental sodium reduction strategies.”

27. Line 31: “Other studies comparing the sodium contents against the country-specific sodium showed higher proportion with about half of meat products meeting the individual targets in Australia, South Africa and the US.11 12 20.It was therefore worth learning for China to take into account both the technical feasibility and consumer acceptability if sodium targets were to be set in the future.”

Please rewrite these sentences, incorrect English language.

Response: Thank you for your valuable advice. These sentences have been re-written and now read as follows: “Studies have shown that in Australia, South Africa and the US, about half of meat products met their own national targets.11 12 20 In summary, the sodium lowering targets provide a level playing field within a country. Many food manufacturers are trying to work towards the targets. This finding also indicates that technical issues should not be a barrier for manufacturers to reformulate their foods.”

28. There were big variations within and between the five countries with different sodium reduction policies, which implies great potential of sodium reduction in meat and fish products by setting feasible sodium reduction targets in countries without sodium reduction program and sustaining robust implementation and monitoring of the targets in the countries with sodium targets in place, as well as selection of less salted food by consumers.

Please rewrite this sentence to be more grammatically correct.

Response: Thank you for your valuable advice. Now we have modified these sentences to “There are large differences in sodium levels of packaged foods among the five countries with different sodium reduction policies. This implies that the target-based strategy is effective in lowering sodium levels in foods. Therefore, setting feasible or further lower sodium targets is urgent. Regular evaluation is also needed to ensure its robust implementation.”

Reviewer 2 grammatical problems:

General response: Thank you for correcting the grammatical problems of this paper. We have accepted all the changes you suggested and already highlighted them in the article.

1. Page 5, line 39-43

“However, the high sodium content, which is known to be a key factor for the quality and sensory attributes of processed meat and fish, is usually of high health concern.”

remove “usually- almost” in all cases

Change to “However, the high sodium content, which is known to be a key factor for quality and sensory attributes of processed meat and fish, otherwise raises a huge public health concern.”

2. Page 5, line 55-60

Although in developing countries like China, sodium intake mainly derives from cooking, the consumption of processed foods including meat and fish products tends to increase with the rapid urbanization and nutrition transition”

Change to “In developing countries like China, sodium intake mainly derives from cooking, yet with the rapid urbanization and dietary transition, the consumption of hidden sodium in processed foods including meat and fish products tends to be increasing rapidly.”

3. Page 6 line 33

A case in point is the sodium content of sauces in China vs UK.

Change to “which can be demonstrated by one in UK vs China: the median sodium content were on average 4.4-fold less in UK sauces compared with their Chinese equivalents.”

4. Page 8 line 44

and expressed as green, amber and red accordingly in a horizontal bar chart to show the sodium contents visually.

Replace “accordingly in” with “accordingly to”

5. Page 9 line 47

The number of products per category ranged from 1 in meat-free products, kebabs and roasted chicken to 2817 in sausages and hot dogs.

Change “meat-free products” to “meat-free alternative products”

Response: As you also mentioned in Question 9, we have changed it to “meat alternative products”.

6. Page 12 line 19-21

“Of the 13 categories of processed meat and fish products with 2017 UK sodium reduction targets,”

Change to “In the 13 categories of processed meat and fish products, ”

7. Page 16, line 36-42

The other four countries, the UK, the US, Australia and South Africa, all have set voluntary or mandatory sodium targets for meat and fish products along within a comprehensive sodium reduction policy/program in these countries.

Change to “The remaining four countries, all have set voluntary or mandatory sodium targets for meat and fish products along with a comprehensive sodium reduction policies/programs in these countries.”

8. Page 18 line 33-35

There were big variations within and between the five countries with different sodium reduction...

Replace “big” with “large”

Reviewer2: Question answers:

9. “Comparisons of sodium content in processed meat and fish products among five countries — potential for reformulation”—potential for feasible targets or formulation?- this study does not make any suggestion as to aspects of reformulation so the title is not appropriate? it would also be good to mention meat alternative products more in depth.

Response: Many thanks. Also referring to the suggestion of the editor, we have changed the title to “Cross-sectional comparisons of sodium content in processed meat and fish products among five countries - potential for feasible targets and reformulation”. Including some discussion for meat

alternatives is a little bit too abrupt. We only share some considerations here for your reference and hope you are satisfied. First, all the five countries adopt the same food categorization system in which meat alternative products are categorized into meat products. Second, although not real meat products, meat alternatives have been produced to replace certain meat products with similar flavor and nutrition. Third, treated as meat replacement, meat alternatives usually adopt similar processing technology and have similar sodium content to real meat products.

10. Page 5 line 8-13:

Strengths and limitations of this study. Strengths: 1) This study is the first time to conduct a cross-sectional survey of the sodium content of processed meat and fish products in supermarkets among five countries using global food composition database.

Is this correct? see reference 13

Response: We used the "first" for two reasons. (1) This is the first time to compare the sodium content of processed meat and fish products. (2) Our study covers more countries, including the UK, Australia, the US, South Africa and China. The study cited by reference 13 was also conducted by our team and only compared the salt content of sauces between UK and China.

11. Page 8, line 3-5

Processed fish products were divided into 4 subcategories, including canned fish, chilled fish, frozen fish and other fish.

How about smoked?

Response: The food classification is based on the Australian Food Switch food classification system, which mainly classifies processed fish products into four categories: canned fish, chilled fish, frozen fish and other fish. Smoked fish is included in chilled fish which contains chilled smoked salmon, chilled raw fish unflavoured, chilled raw fish flavoured, and chilled shellfish.

12. Page 8 line 8-15

Data Exclusion Criteria: Products with no declaration of neither sodium nor salt values were excluded. For identical products with same sodium content in different package sizes, it is regarded as a duplicate product, only one product was included.

Why? this does not make sense- your data should have been analyzed per 100 g? so there is no need for this statement?

Response: Your question makes sense. However, in the process of data cleaning, we found that for some cases, the same product with different package sizes had different bar codes. It is not appropriate to classify it as two goods. Therefore, we included this situation in the data exclusion criteria.

13. Page 11 line 27

For example, the sodium content of roasted chicken in China was 4.5 times that of the UK (893 mg/100g vs 197 mg/100g); chilled fish in China, 4.5 times that of the US (1744 mg/100g vs 389 mg/100g); pate and meat spreads in China, about 4 times that of Australia (1916 mg/100g vs 480 mg/100g).

Where are the p values?

Response: Thanks. we have added the p values and it now reads: "For example, the sodium content of roasted chicken in China was 4.5 times that of the UK (893 mg/100g vs 197mg/100g) ($p < 0.001$); chilled fish in China, 4.5 times that of the US (1744 mg/100g vs 389 mg/100g) ($p < 0.001$); pate and meat spreads in China, about 4 times that of Australia (1916 mg/100g vs 480mg/100g) ($p < 0.001$)"

14. Page 11 line 36

However, the sodium content of bacon, frozen meat, salami and cured meats, dried meat and frozen fish in China was the lowest among five countries.

This could have been illustrated in a figure?

Response: Thank you for your suggestion. We have added Figure 2-1, 2-2, 2-3, 2-4, and 2-5 to illustrate the data.

15. Page 12 line 33-38

The UK had the highest percentage of achieving the targets except in the bacon category, only 14.0% of bacon in the UK reached the target, lower than the US (28.2%), Australia (50.2%), SA (75.0%) and China (84.9%).

Smoked or un-smoked?

Response: Thank you for your question. In the Australian Food Switch classification system, bacon has no further categorization and it covers all kinds of bacon, including smoked and un-smoked.

16. Page 14 line 10

Table 3 Contribution (%)¹ towards the WHO daily intake recommendation (2000 mg/d) for each serving (100g) consumption of processed meat and fish products

Font is not consistent

Response: Thank you for your comment. Also referring to Reviewer 1's comments, we have re-written the title of Table 3 and the corresponding parts in Methods, Results and the note to Table 3. The font is also consistent now.

17. Page 15 line 11

The results showed extremely wide discrepancy within and between countries.

Why is this surprising? considering the difference in dietary habits? this data would be more meaningful if it was evaluated or correlated with the prevalence of Hypertension or its' annual incidents within an between countries

Response: Thank you for suggestions for further studies. We also assume that the difference in sodium contents of packaged meat and fish products is large among different countries considering the difference in culture and dietary habits, but the extent of the difference is not clear. Our findings confirmed us the large difference specifically and it is of great significance to instruct food product reformulation and development of salt reduction policies. At present, the available data cannot support us to analyze the relationship between food sodium content and the prevalence of hypertension. But, we agree that your suggestion is really a good idea for further research and should be doable.

18. Page 15 line 16-23

The sodium content of meat and fish products in each country was high compared with "Traffic Light" criteria with only 10% of the products falling into the green light group in the UK and US and no more than 5% in China and South Africa.

So do authors believe traffic light is a helpful measure for regulating products? unfortunately consumer purchase data is not available to assess this.

Response: Yes, we have no consumer purchase data to show the effect of Traffic Light, although some studies have shown that traffic light labelling helps guide consumers to choose healthier foods. [Temple NJ. Front-of-package food labels: A narrative review. *Appetite*. 2020;144:104485. PMID: 31605724.]. In this study, we used the Traffic Light criteria mainly for comparison purpose.

19. Page 15 line 23-25

The percentage of products meeting 2017 UK sodium reduction targets were generally low ranging from 7.1% (China) to 26.6% (the UK).

This again is not surprising...

Response: Yes, we agree, but we do hope this finding could remind the potential of sodium reduction through progressing sodium target setting. This could be more meaningful for countries with lower target achievement.

20. Page 15 line 43-50

In China, where 70%~80% of sodium came from cooking at home, with a remarkable increase in consumption of processed foods and meals out of home in recent years, sodium intake from meat and fish products is an emerging concern.

If these findings and data are already in scientific literature, it is difficult to see what this study is contributing to the body of knowledge? Authors need to highlight the novelty here, if any?

Response: With development and urbanization, more and more countries have realized the increasing challenge of pre-packaged food to health. The findings in this study could be a reminder for specific strategies to promote sodium reduction by thinking about the following questions. What the gap and space is for a country in sodium reduction for pre-packaged food? Which products should be targeted on first? Whether and how to adopt the target setting strategy, mandatory or voluntary? And how to overcome the barriers from manufacturers who may be reluctant to reformulate their product by arguing that salt reduction would shorten the shelf life? So, although not surprising to many people, the specific findings are still of significance for governments and other stakeholders to take steps in setting sodium targets and reformulation for pre-packaged products. This paragraph has been revised and added in the Discussion section ahead of strength and limitation part.

21. Page 15 line 57-60

One strategy to reduce sodium intake from meat and fish products would be to replace high-sodium products with low-sodium products.

Is it this simple? how about consideration such as flavor, texture also safety?

Response: We are recommending this strategy for consumers and this has been made clear by changing the sentence to "One strategy to reduce sodium intake from packaged products is to encourage consumers to replace high-sodium products with low-sodium products."

From the perspective of manufacturers, it is not easy to simply replace or reformulate the high-sodium products which already exists for years. However, the large difference in sodium content of similar products in different countries, and the difference in sodium content among different brands within the same country, indicate that there is still a lot of room for salt reduction. Product features regarding satisfying flavor, texture, safety and stability have been the key considerations for manufacturers, but three other situations should also be considered. First, product formulations might have been lagged behind consumers' requirement for less sodium products. Second, many manufacturers may resist reformulation due to unfounded concern for flavor acceptance and safety. Third, a 10-15% reduction in sodium will go undetected, and the product reformulation could be done step by step. This paragraph has also been included in the Discussion section.

22. Page 16 line 3-6

For example, choosing raw unflavoured meats instead of salami and cured meats would decrease ten to twenty times of sodium intake in all countries.

but this is about consumer choice rather than reformulation?

Response: You are right. We encourage consumers to choose lower-salt products. To make this clear, we add "encourage consumers to..." in the preceding sentence as mentioned in the answers to question 21 above.

23. Page 16 line 13-15

Therefore, reducing sodium in all meat and fish products would be the optimal strategy.

Again, nothing new here

Response: We have removed this sentence and replaced it by encouraging the development of new products with the same or better flavor and less sodium.

24. Page 16, line 20-26

Additionally, the comparison of sodium contents across the countries with different sodium reduction policies in meat and fish products suggested that setting sodium targets for processed foods would be

an effective way to reduce sodium contents of packaged foods, which is in alignment with many other studies.

This is the only valuable conclusion in this study...

Response: Thanks, we have moved this statement to the first sentence of the paragraph and followed by the discussion on this topic for each of the five countries.

25. Page 16, line 36-42

The other four countries, the UK, the US, Australia and South Africa, all have set voluntary or mandatory sodium targets for meat and fish products along within a comprehensive sodium reduction policy/program in these countries.

So UK is not the only country with targets? yet all these countries vary significantly? so in reality, target is not end and all?

Response: Yes, UK is not the only country with targets. As mentioned in the discussion, China has no sodium targets to limit the sodium added to the products. The UK has issued four sets of voluntary sodium targets. The US and Australia set the voluntary sodium targets for various processed foods. South Africa is the first country to regulate legislated sodium limits for a range of food products in 2012.

Yes, the sodium contents also vary significantly across the four countries with targets, but all had significant lower sodium content than China. We admit target is not end and all, it is only a starting point. The paragraph initiated with "Target implementation is also critical" has explained the variation of target achievement among different product categories and among different countries, indicating the importance of target implementation.

26. Page 18 line 17-24

Although the crowdsourcing element of the data collection may in part reflect what consumers are eating, future studies should consider using proper product sales data or consumption data to estimate the actual sodium intake from processed meat and fish products in each country.

it will only be useful to see the traffic system is effective in helping the consumer making a healthier choice... perhaps only in case of UK. The authors are not making any reference to whether an equivalent system exist in other countries?

Response: In this study, the food labeling information has been mainly collected by trained data collectors in middle-large supermarkets. It has no relationship with food consumption or sales. Only a small part of data is collected through "crowdsourcing" by consumers during shopping. That's way we admit the limitation that "we did not capture food-purchasing data to quantify actual sodium consumption of processed meat and fish products, although the crowdsourcing element of the data collection may somehow reflect what consumers have eaten. Future studies should consider using more reliable product sales data or consumption data to estimate the actual sodium intake from processed meat and fish products in each country." So, it has little relationship with Traffic Lights labeling.

As far as we know, there are dozens of countries have been using similar healthy choice like "Traffic Lights" to guide consumers for healthier food choice. Food Switch also has built-in electronic Traffic Light and Healthy Star labeling.

27. Page 33-47

There were big variations within and between the five countries with different sodium reduction policies, which implies great potential of sodium reduction in meat and fish products by setting feasible sodium reduction targets in countries without sodium reduction program and sustaining robust implementation and monitoring of the targets in the countries with sodium targets in place, as well as selection of less salted food by consumers.

This sentence is too long....

Response: Thank you for your suggestion. We will change it to "There are large differences in sodium levels of packaged foods among the five countries with different sodium reduction policies. This

implies that the target-based strategy is effective in lowering sodium levels in foods. Therefore, setting feasible or further lower sodium targets is urgent. Regular evaluation is also needed to ensure its robust implementation. ”

VERSION 2 – REVIEW

REVIEWER	Charlton, Karen University of Wollongong, School of Health Sciences
REVIEW RETURNED	14-Mar-2021
GENERAL COMMENTS	All requested changes have been made satisfactorily. The article has been much improved and now warrants publication.